# Unfolding Taylor's Approximations for Image Restoration

**Man Zhou**[1]    **Xueyang Fu** [1][*]   **Zeyu Xiao**[1]    **Aiping Liu** [1]    **Gang Yang**[1]    **Zhiwei Xiong** [1]

[1]University of Science and Technology of China

{manman}@mail.ustc.edu.cn   xyfu@ustc.edu.cn

## Abstract

Deep learning provides a new avenue for image restoration, which demands a delicate balance between fine-grained details and high-level contextualized information during recovering the latent clear image. In practice, however, existing methods empirically construct encapsulated end-to-end mapping networks without deepening into the rationality, and neglect the intrinsic prior knowledge of restoration task. To solve the above problems, inspired by *Taylor's Approximations*, we unfold Taylor's Formula to construct a novel framework for image restoration. We find the main part and the derivative part of Taylor's Approximations take the same effect as the two competing goals of high-level contextualized information and spatial details of image restoration respectively. Specifically, our framework consists of two steps, correspondingly responsible for the mapping and derivative functions. The former first learns the high-level contextualized information and the later combines it with the degraded input to progressively recover local high-order spatial details. Our proposed framework is orthogonal to existing methods and thus can be easily integrated with them for further improvement, and extensive experiments demonstrate the effectiveness and scalability of our proposed framework. Code will be publicly available upon acceptance.

## 1   Introduction

Image restoration has long been an important task in computer vision, which demands a delicate balance between spatial fine-grained details and high-level contextualized information while recovering a latent clear image from a given degraded observation. It is a highly ill-posed issue as there exists infinite feasible results for single degraded image. Representative image restoration tasks include image deraining, image deblurring.

Much research efforts have been devoted to solve the single image restoration problem, which can be categorized into two groups: traditional optimization methods [16, 36] and deep learning based methods. In detail, various natural images priors have been developed in traditional image restoration methods to regularize the solution space of the latent clear image, *e.g.*, low-rank prior [36, 42], dark channel prior [31, 32, 54], graph-based prior [28, 2], total variation regularization [5, 8, 1] and sparse image priors [29, 56]. However, these priors are difficult to design and the methods are difficult to optimize, limiting their practical usage.

Convolutional neural networks (CNNs) have been applied in numerous high-level computer vision problems and obtained promising improvement in image restoration tasks over traditional methods [52, 63, 15, 26, 45, 25, 7, 66, 67, 10, 37]. However, most existing CNN-based image restoration methods empirically construct encapsulated end-to-end mapping networks without deepening into the rationality, and neglect the intrinsic properties of the degradation process. Specifically, it makes

---

[*]Corresponding Author.

35th Conference on Neural Information Processing Systems (NeurIPS 2021).

these methods unable to balance effectively the two competing goals of spatial fine-grained details and high-level contextualized information while recovering images from the degraded inputs, limiting the model performance. Moreover, existing forward-pass mapping methods lack of sufficient interpretability, making it hard to analyze the effect of each module and preventing further improvement.

To solve the aforementioned problems, in this paper, we first revisit the connection between Taylor's Approximation and image restoration task, and propose to unfold Taylor's Formula as blueprints to construct a novel framework that enforces each process part to coincide with the intrinsic properties of image restoration task. We figure out that the main part and derivative part of Taylor's Approximation take the same effect as the above two competing goals of image restoration respectively. In this way, our approach deviates from existing methods that optimally balance the above goals in the overall recovery process and independent from the degradation process. Specifically, we break down the overall recovery process into two manageable steps, corresponding to two specific operation steps that are responsible for the mapping and derivative functions respectively. The former first learns the high-level contextualized information and the later combines it with the degraded input to progressively recover local high-order spatial details. Moreover, our proposed framework is orthogonal to existing methods and thus can be easily integrated with them for further performance gain. The conducted extensive experiments demonstrate the effectiveness and scalability of our proposed framework over two popular image restoration tasks, image deraining and image deblurring.

The contributions of this paper can be summarized as follows:

1) We introduce a new perspective for designing image restoration framework by unfolding the Taylor's Formula. To the best of our knowledge, this is the first effort to solve image restoration task inspired by Taylor's Approximations.
2) In contrast to existing methods that optimally balance the competing goals of high-level contextualized information and local high-order spatial details of image restoration respectively in the overall recovery process, we break down it into two manageable steps, corresponding to two specific steps that are responsible for the mapping and derivative functions respectively.
3) Our proposed framework (*i.e.*, Deep Taylor's Approximations Framework) is orthogonal to existing CNN-based methods and these methods can be easily integrated with our framework directly for further improvement. Extensive experiments conducted on standard benchmarks demonstrate the effectiveness of the framework.

## 2  Related Work

Image restoration aims to recover the latent clean images from its degraded observation, which is beneficial for several downstream high-level computer vision tasks, *e.g.*, object detection, image classification and image segmentation. It is a highly ill-posed problem as there exists infinite feasible results for a single degraded image. To this end, it attracts more attention over the whole community. Representative tasks of different degradation include image deraining, image deblurring.

**Image deraining** aims to remove the rain streaks while maintaining its image textures, and has gained much advances owing to the breakthrough of deep learning [12, 59, 60, 13, 48, 27, 28, 35, 47, 9, 68]. Fu *et al.* [11] early attempt to design a CNN-based architecture for image deraining and obtain favorable performance with a larger margin than classic promising methods such as guided filter, nonlocal means filtering and more [50, 3, 22, 34]. Yang *et al.* [55] introduce a multi-task network with a series of contextualized dilated convolution and recurrent framework to achieve joint detection and removal of rain streaks. Ren *et al.* [35] present the PreNet as a simple baseline for image deraining that combines Resblocks and recurrent layers in a progressive and recursive manner. Overall, the state-of-the-art deep learning methods rely on designing complex network structure. However, these complicated network architectures make them lack of evident interpretability and still have room for further improvement.

**Image deblurring** is a typical ill-posed problem which aims at generating a sharp latent image from a blurry observation. Early Bayesian-based iterative deblurring methods include the Wiener filter [49] and the Richardson-Lucy algorithm [38]. Later works commonly rely on developing effective image priors [24, 40, 51, 69] or sophisticated data terms [6]. Recently, several CNN-based methods have been proposed for image deblurring [17, 25, 30, 63, 26, 45, 41, 15, 64, 33, 61, 43, 37]. For example,

Sun *et al.* [44] propose a CNN-based model to estimate a kernel and remove non-uniform motion blur. Chakrabarti [4] uses a network to compute estimations of sharp images that are blurred by an unknown motion kernel. Nah *et al.* [30] propose a multi-scale loss function to apply a coarse-to-fine strategy. Kupyn *et al.* propose DeblurGAN [25] and DeblurGAN-v2 [26] to remove blur based on adversarial learning. Despite of the encouraging performance achieved by CNN-based methods for image deblurring, they fail to reconstruct sharp results with good interpretability.

**The Taylor's Approximations** is one of the most popular methods in numerical approximation. In previous researches, Tukey [46] is the first to exploit the Taylor's series expansion to study error propagation. Afterwards, a number of studies [18, 20, 21, 23, 19] have been made and based on the first-order Taylor's expansion. It is, after all, a liner approximations that results in the inaccurate values. Xue *et al.* [53] develop above methods by high-order Taylor series expansion, which further improve the performance . However, it has never been explored for image restoration. In this paper, we revisit the connection between the Taylor's Approximations and image restoration process for the first time, and propose to unfold the Taylor's Formula to construct a novel framework, named Deep Taylor's Approximations Framework.

## 3 Proposed Method

### 3.1 Revisiting Taylor's Approximations and Image Restoration

**Problem Formulation** Image degradation is the most common and inevitable phenomena in imaging systems. It is commonly formulated as

$$y = Ax + N,\tag{1}$$

where $y$, $x$, $A$ and $N$ denote a degraded observation, a latent clear image, a degraded matrix and noise respectively. Specifically, when $A$ is the identity matrix and $N$ is the rain streak, it transforms as image deraining task. Moreover, when $A$ is the blurry matrix and $N$ is the additional noise, it turns to image deblurring task. Observing Equation (1), for the inverse process, it can be reformulated as

$$y_0 = Ax = (y - N),\tag{2}$$

where we denote the term $Ax$ as $y_0$, representing the latent image without additional noise. In detail, $N$ is rain streak $R$ over image deraining task while acting as the additional noise in image deblurring.

**Existing Methods** Most of the existing CNN-based image restoration methods empirically construct encapsulated end-to-end mapping networks which neglect the intrinsic prior knowledge of image restoration task. Specifically, it acts as

$$x = F(y),\tag{3}$$

where $F$ is the existing-designed mapping network, which empirically learns the mapping between the degraded and clean pairs $x$ and $y$.

**Ours: Revisiting Image Restoration Task** Different from existing methods, we try to associate Taylor's Approximations with image restoration, which has not been considered in previous methods. Referring to Equation (2), we define the inverse process using the function $F$, and then we can obtain the relationship as

$$x = F(y_0) = F(y - N).\tag{4}$$

It can be figured out that the mapping function $F$ in our framework corresponds the inverse operation of degradation matrix $A$. Deviating from most existing CNN-based methods, our framework considers the intrinsic prior knowledge of image restoration task and has better interpretability. The inverse operation of degradation matrix $A$ is denoted as $A^-$

$$F = A^-.\tag{5}$$

Let $-\boldsymbol{N} = \boldsymbol{\epsilon} = \boldsymbol{y}_0 - \boldsymbol{y}$, we expand Equation (4) with an infinite-order Taylor's series expansion of a scalar-valued of more than one variable, written compactly as

$$\boldsymbol{x} = \boldsymbol{F}(\boldsymbol{y}_0) = \boldsymbol{F}(\boldsymbol{y} + \boldsymbol{\epsilon}) \tag{6}$$

$$= \boldsymbol{F}(\boldsymbol{y}) + \frac{1}{1!}\sum_{i=1}^{n}\frac{\partial \boldsymbol{F}(\boldsymbol{y})}{\partial y_i}\boldsymbol{\epsilon}_i + ... + \frac{1}{k!}\sum_{i_1,...,\,i_k\in\{1,2,...,n\}}\frac{\partial^k \boldsymbol{F}(\boldsymbol{y})}{\partial y_{i_1}...\partial y_{i_k}}\boldsymbol{\epsilon}_{i_1}...\boldsymbol{\epsilon}_{i_k} + ... \tag{7}$$

$$= \sum_{k=0}^{\infty}\frac{1}{k!}\left(\sum_{i=1}^{n}\boldsymbol{\epsilon}_i\frac{\partial}{\partial y_i}\right)^k \boldsymbol{F}(\boldsymbol{y}), \tag{8}$$

where we use the notation

$$\left(\sum_{i=1}^{n}\boldsymbol{\epsilon}_i\frac{\partial}{\partial y_i}\right)^k \boldsymbol{F}(\boldsymbol{y}) \equiv \sum_{i_1=1}^{n}\sum_{i_2=1}^{n}...\sum_{i_k=1}^{n}\frac{\partial^k \boldsymbol{F}(\boldsymbol{y})}{\partial y_{i_1}...\partial y_{i_k}}\boldsymbol{\epsilon}_{i_1}...\boldsymbol{\epsilon}_{i_k} \tag{9}$$

$$= \sum_{i_1,...,\,i_k\in\{1,2,...,n\}}\frac{\partial^k \boldsymbol{F}(\boldsymbol{y})}{\partial y_{i_1}...\partial y_{i_k}}\boldsymbol{\epsilon}_{i_1}...\boldsymbol{\epsilon}_{i_k}. \tag{10}$$

When only regarding $n$ order Taylor's Approximations, it can be simplified as

$$\boldsymbol{x} = \boldsymbol{F}(\boldsymbol{y}) + \sum_{k=1}^{n}\frac{1}{k!}\left(\sum_{i=1}^{n}\boldsymbol{\epsilon}_i\frac{\partial}{\partial y_i}\right)^k \boldsymbol{F}(\boldsymbol{y}). \tag{11}$$

It can be separated into two parts for consideration. In detail, the first term, as main part, $\boldsymbol{F}(\boldsymbol{y})$ represents the high-level contextualized information, which gives a constant approximation of clear image (ground truth) while the rest is the local high-order spatial details. When we revisit the connection between the goals of image restoration and Equation (11), we can clearly find that the main part and derivative part of Taylor's Approximations take the same effect as the two competing goals of high-level contextualized information and spatial details of image restoration respectively. In this way, our approach deviates from existing methods that optimally balance the above goals in the overall recovery process and independent from the degradation process. Specifically, we break down the overall recovery process into two manageable steps. The former first learns the high-level contextualized structures and the later combines them with the degraded input to progressively recover local high-order spatial details. The details of our proposed framework are illustrated as bellow.

### 3.2 Deep Taylor's Approximations Framework

Based on above analysis, we reasonably associate Taylor's Approximations with image restoration task, which has not been considered in previous restoration methods. In detail, we propose to unfold Taylor's Formula as blueprints to construct a novel framework, named Deep Taylor's Approximations Framework. In this way, almost every process part one-to-one corresponds to each operation involved in Taylor's Formula, thereby breaking down the overall recovery process into two manageable steps. Specifically, our framework consists of two operation parts, correspondingly responsible for the mapping and derivative functions.

**Mapping Function Part** As aforementioned function $\boldsymbol{F}$, it takes responsibility for mapping the degraded input to the approximation of expected latent clear image. As shown in Figure 1, it can be implemented with existing CNN-Based or traditional methods. In the following, we choose several representative image restoration methods as $\boldsymbol{F}$ to validate the effectiveness and scalability.

**Derivative Function Part** Recalling Equation (11), for the $k$ order derivative part, it can be written as

$$\boldsymbol{F}^{(k)}(\boldsymbol{\epsilon})(\boldsymbol{\epsilon})^k = \sum_{i_1,...,\,i_k\in\{1,2,...,n\}}\frac{\partial^k \boldsymbol{F}(\boldsymbol{y})}{\partial y_{i_1}...\partial y_{i_k}}\boldsymbol{\epsilon}_{i_1}...\boldsymbol{\epsilon}_{i_k}, \tag{12}$$

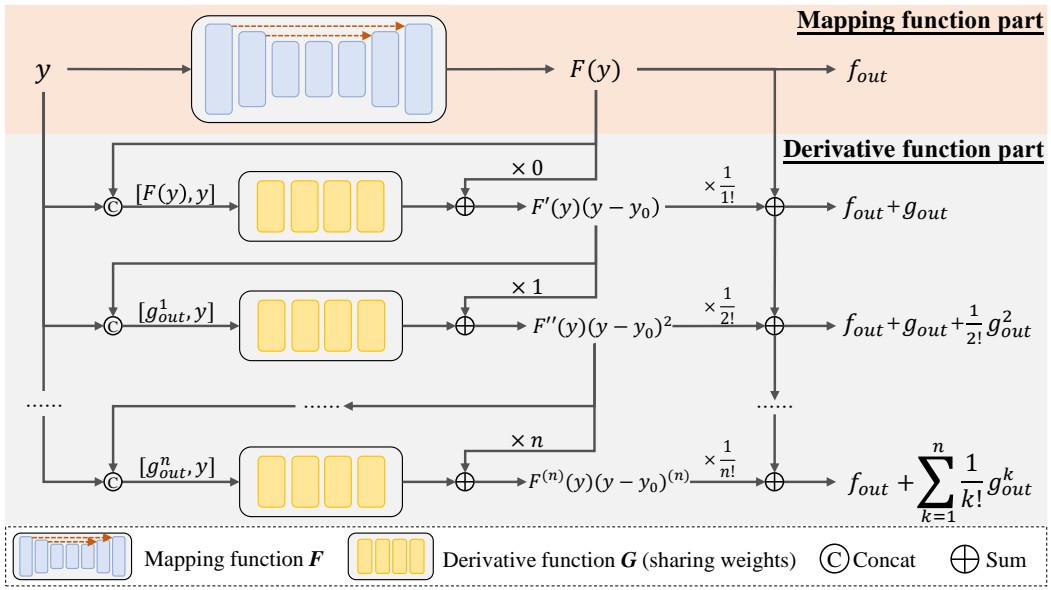

**Figure 1:** The overview structure of Deep Taylor's Approximations Framework. It consists of two parts, Mapping function part $\boldsymbol{F}$ and Derivative function part $\boldsymbol{G}$. The former learns to map the degraded input image $y$ to the main energy of ground truth, while the later combines them with the degraded input to progressively recover the local high-order details. The parameters of Derivative function part $\boldsymbol{G}$ are shared across the progressive stages. The whole framework is trained in an end-to-end manner.

differentiating above $k$ order part $\boldsymbol{F}^{(k)}(\boldsymbol{y})(\boldsymbol{\epsilon})^k$ for $y$ as

$$\frac{\partial \boldsymbol{F}^{(k)}(\boldsymbol{y})(\boldsymbol{\epsilon})^k}{\partial \boldsymbol{y}} = \boldsymbol{F}^{(k+1)}(\boldsymbol{y})(\boldsymbol{\epsilon})^k - \boldsymbol{F}^{(k)}(\boldsymbol{y}) \times k(\boldsymbol{\epsilon})^{k-1}, \tag{13}$$

multiplying above Equation (13) by $\epsilon$ as

$$\frac{\partial \boldsymbol{F}^{(k)}(\boldsymbol{y})(\boldsymbol{\epsilon})^k}{\partial \boldsymbol{y}} \times \boldsymbol{\epsilon} = (\boldsymbol{F}^{(k+1)}(\boldsymbol{y})(\boldsymbol{\epsilon})^k - k\boldsymbol{F}^{(k)}(\boldsymbol{y})(\boldsymbol{\epsilon})^{k-1}) \times \boldsymbol{\epsilon} \tag{14}$$

$$= \boldsymbol{F}^{(k+1)}(\boldsymbol{y})(\boldsymbol{\epsilon})^{k+1} - k\boldsymbol{F}^{(k)}(\boldsymbol{y})(\boldsymbol{\epsilon})^k. \tag{15}$$

To this end, we exploit Derivative function sub-network, named $\boldsymbol{G}$ to take effect as above process. We denote the $k$ order output of network $\boldsymbol{G}$ as $\boldsymbol{F}^{(k)}(\boldsymbol{y})(\boldsymbol{\epsilon})^k$, simply recorded as $g_{out}^k$. Referring Equation (15), we can find out the connection between $k$ order output and $k+1$ order one

$$g_{out}^{k+1} = \boldsymbol{G}(g_{out}^k) + k\boldsymbol{F}^{(k)}(\boldsymbol{y})(\boldsymbol{\epsilon})^k. \tag{16}$$

Replacing $\boldsymbol{F}^{(k)}(\boldsymbol{y})(\boldsymbol{\epsilon})^k$ with $g_{out}^k$ as

$$g_{out}^{k+1} = \boldsymbol{G}(g_{out}^k) + k \cdot g_{out}^k. \tag{17}$$

**Implementation**  Referring above analysis, we can find that the Mapping function part $\boldsymbol{F}$ only requires the degraded image as input as

$$f_{out} = \boldsymbol{F}(\boldsymbol{y}), \tag{18}$$

referring Equation (17), the Derivative function part $\boldsymbol{G}$ needs the $g_{out}^k$ from the Mapping function sub-network $\boldsymbol{F}$ and blurry image $y$. It is because that the unfolding iteration process of $\boldsymbol{G}$ running involves $y$. In this regard, concatenating $g_{out}^k$ and $y$ into $\boldsymbol{G}$ as input for inference

$$g_{out}^{k+1} = \boldsymbol{G}(Concat([g_{out}^k, \boldsymbol{y}])). \tag{19}$$

Taking together above two operation steps, the final output of $n$ order Deep Taylor's Approximations framework can be obtained as

$$O = f_{out} + \sum_{k=1}^{n} \frac{1}{k!} g_{out}^k. \tag{20}$$

**Loss Function** Following the setting in [14], we apply the $\mathcal{L}_1$ loss function to optimize the proposed framework. Suppose $x$ denotes the corresponding ground truth of given degraded image $y$, the final loss can be recorded as

$$L = \mathcal{L}_1(x - O) + \lambda\mathcal{L}_1(x - f_{out}), \tag{21}$$

where $\lambda$ is a weighted factor.

## 4 Experiments

To demonstrate the effectiveness and scalability of our proposed framework, we conduct experiments over two image restoration tasks, *i.e.*, image deraining and image deblurring. In the following part, we first conduct the ablation experiments over several representative methods with/without integrating with our framework and report results on standard benchmarks. And then, we show the quantitative and qualitative results to explore our Deep Taylor's Approximations Framework with different series orders. Finally, we demonstrate the visual results to further validate the interpretability and the underlying mechanism.

### 4.1 Experimental Settings

For image deraining task, we choose four representative deraining methods, including DDN [12], PReNet [35], DCM [14] and MPRnet [57] to integrate with our proposed Deep Taylor's Approximations Framework for comparison between the integrated and the original. Specifically, all the aforementioned methods are exploited as mapping function part of our framework and then operate together with derivative function part for image deraining. Regarding experiment rainy datasets, the first three methods including DDN, DCM and PReNet are trained and evaluated over three widely-used standard benchmark datasets, including Rain100H, Rain100L, Rain800. In detail, Rain100H, a heavy rainy dataset are composed with 1,800 rainy images for training and 100 rainy samples for testing. However, referring the PReNet work, we find that 546 rainy images from the 1,800 training samples have the same background contents with testing images. Therefore, for fair comparison, we screen out these 546 rainy images from training set, and train all the selected baseline models on the remaining 1,254 training images. And, Rain100L dataset contains 200 training samples and 100 testing images with light level degraded rain streaks. In addition, the Rain800 is proposed in [58], which includes 700 training and 100 testing images. As for the recent method MPRnet, referring original paper, 13,712 clean-rain image pairs gathered from Rain100H, Rain100L, Test100, Test2800 and Test1200 are used for training and two above Rain100H, Rain100L for testing. For fair comparison, we follow the datasets setting as original paper. For the implementation, owing to unreleased open codes of DCM, we insert it into the training framework of PreNet for experiment comparison while the remaining follows the same pipeline.

For image deblurring task, typical methods like DCM [14], MSCNN [30] and RDN [66, 65] are adopted in our experiments. As in [62, 45, 26], we use the GoPro [30] dataset that contains 2,103 image pairs for training and 1,111 pairs for evaluation. Furthermore, to demonstrate generalizability, we take the model trained on the GoPro dataset and directly apply it on the test images of the HIDE [41] and RealBlur [39] datasets. The HIDE dataset is specifically collected for human-aware motion deblurring and its test set contains 2,025 images. While the GoPro and HIDE datasets are synthetically generated, the image pairs of RealBlur dataset are captured in real-world conditions. The RealBlur dataset has two subsets: (1) RealBlur-J is formed with the camera JPEG outputs, and (2) RealBlur-R is generated offline by applying white balance, demosaicking, and denoising operations to the RAW images.

For quantitative comparisons, two popular measurement metrics are employed in the following quantitative comparison, the peak signal-to-noise ratio (PSNR) and structural similarity index (SSIM).

### 4.2 Integrating Existing Methods into Our Framework

We evaluate the integrated models against their baselines (*i.e.*, the models trained without integration) in terms of PSNR/SSIM with 3-order (see Section 4.3). As shown in Table 1 and Table 2, we can clearly find that, by integrating with our proposed framework, all the baselines obtain performance gain over all the datasets in image deraining and image deblurring task, which validates the effectiveness of our framework. For example, in Table 2, DCM [14] obtains 0.36dB and 0.14dB psnr

gain on GoPro and RealBlur dataset. Taking two representative examples in Figure 3, the result of "Integrated" is cleaner than that of "Original" with fewer blurry effect. Meanwhile, for image deraining, the results of "Original" maintain the better spatial details than that of "Integrated". The quantitative comparison from Table 1 also testifies above analysis.

**Table 1:** Comparison of quantitative results in terms of PSNR (dB) and SSIM over image deraining. "Original" represents to employ the same architecture as original paper, and "Integrated" indicates to integrate them with our proposed framework. The corresponding experiment setting can be referred as section 4.2.

| Model | Methods | Rain100H | | Rain100L | | Rain800 | |
|---|---|---|---|---|---|---|---|
| | | PSNR | SSIM | PSNR | SSIM | PSNR | SSIM |
| DDN [12] | Original | 22.26 | 0.690 | 34.85 | 0.951 | 24.04 | 0.867 |
| | Integrated | 22.34 | 0.701 | 35.16 | 0.953 | 24.21 | 0.867 |
| PReNet [35] | Original | 26.77 | 0.858 | 32.44 | 0.950 | 22.03 | 0.720 |
| | Integrated | 26.78 | 0.858 | 34.44 | 0.950 | 22.04 | 0.723 |
| DCM [14] | Original | 28.66 | 0.889 | 37.15 | 0.980 | 26.78 | 0.859 |
| | Integrated | 28.77 | 0.901 | 37.31 | 0.981 | 26.93 | 0.861 |
| MPRnet [57] | Original | 30.44 | 0.872 | 36.24 | 0.959 | - | - |
| | Integrated | 30.52 | 0.873 | 36.30 | 0.960 | - | - |

**Table 2:** Comparison of quantitative results in terms of PSNR (dB) and SSIM over image deblurring. All the models follow the same settings as original works.

| Model | Methods | GoPro | | HIDE | | RealBlur-J | | RealBlur-R | |
|---|---|---|---|---|---|---|---|---|---|
| | | PSNR | SSIM | PSNR | SSIM | PSNR | SSIM | PSNR | SSIM |
| DCM [14] | Original | 30.25 | 0.929 | 28.00 | 0.901 | 35.55 | 0.927 | 28.60 | 0.871 |
| | Integrated | 30.39 | 0.933 | 28.10 | 0.926 | 35.91 | 0.930 | 28.69 | 0.873 |
| MSCNN [30] | Original | 29.08 | 0.914 | 25.72 | 0.874 | 32.51 | 0.841 | 27.87 | 0.827 |
| | Integrated | 29.19 | 0.916 | 25.80 | 0.877 | 32.79 | 0.852 | 27.99 | 0.830 |
| RDN [66] | Original | 29.20 | 0.929 | 26.44 | 0.859 | 28.38 | 0.899 | 26.50 | 0.871 |
| | Integrated | 29.31 | 0.934 | 26.50 | 0.866 | 28.44 | 0.901 | 26.56 | 0.880 |

## 4.3 Deep Taylor's Approximations Framework with Different Orders

In this section, we perform the ablation studies about different orders of our proposed Deep Taylor's Approximations Framework. For simplicity, we take the representative method, *i.e.*, DCM, to validate the underlying mechanism over image deraining and image deblurring. In detail, DCM is used for mapping function part and taken together with different orders derivative function part from 0 to 6.

**Table 3:** Comparison of quantitative results in terms of PSNR (dB) and SSIM about the different order Taylor's approximation results from 0 to 6 over image deraining.

| Model | Methods | Rain100H | | Rain100L | | Rain800 | |
|---|---|---|---|---|---|---|---|
| | | PSNR | SSIM | PSNR | SSIM | PSNR | SSIM |
| DCM [14] | order-0 | 28.66 | 0.889 | 37.15 | 0.980 | 26.78 | 0.859 |
| | order-1 | 28.67 | 0.889 | 37.17 | 0.980 | 26.78 | 0.859 |
| | order-2 | 28.71 | 0.890 | 37.20 | 0.980 | 26.79 | 0.859 |
| | order-3 | 28.77 | 0.901 | 37.31 | 0.981 | 26.93 | 0.861 |
| | order-4 | 28.69 | 0.889 | 37.22 | 0.980 | 26.76 | 0.859 |
| | order-5 | 28.73 | 0.889 | 37.24 | 0.981 | 26.83 | 0.860 |
| | order-6 | 28.77 | 0.891 | 37.18 | 0.980 | 26.72 | 0.859 |

**Table 4:** Comparison of quantitative results in terms of PSNR (dB) and SSIM about the different order Taylor's approximation results from 0 to 6 over image deblurring.

| Model | Methods | GoPro | | HIDE | | RealBlur-J | | RealBlur-R | |
|---|---|---|---|---|---|---|---|---|---|
| | | PSNR | SSIM | PSNR | SSIM | PSNR | SSIM | PSNR | SSIM |
| DCM [14] | order-0 | 30.32 | 0.921 | 28.03 | 0.913 | 35.83 | 0.918 | 28.62 | 0.862 |
| | order-1 | 30.33 | 0.922 | 28.05 | 0.914 | 35.85 | 0.918 | 28.64 | 0.862 |
| | order-2 | 30.36 | 0.928 | 28.07 | 0.920 | 35.88 | 0.925 | 28.66 | 0.867 |
| | order-3 | 30.39 | 0.933 | 28.10 | 0.926 | 35.91 | 0.930 | 28.69 | 0.873 |
| | order-4 | 30.35 | 0.932 | 28.06 | 0.925 | 35.87 | 0.929 | 28.65 | 0.872 |
| | order-5 | 30.36 | 0.925 | 28.07 | 0.918 | 35.88 | 0.922 | 28.66 | 0.865 |
| | order-6 | 30.36 | 0.932 | 28.08 | 0.924 | 35.89 | 0.929 | 28.66 | 0.871 |

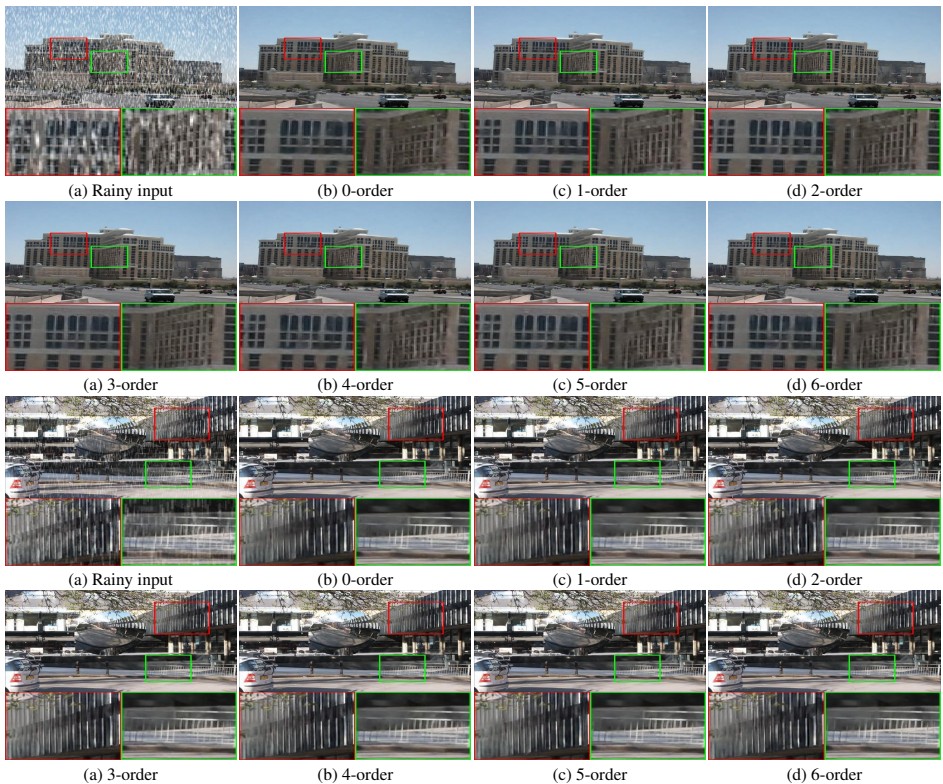

| (a) Rainy input | (b) 0-order | (c) 1-order | (d) 2-order |
| (a) 3-order | (b) 4-order | (c) 5-order | (d) 6-order |
| (a) Rainy input | (b) 0-order | (c) 1-order | (d) 2-order |
| (a) 3-order | (b) 4-order | (c) 5-order | (d) 6-order |

**Figure 2:** Visual results from the Rain800 rainy dataset for DCM [14] with different-order framework.

The 0-order represents the original DCM [14] method. And the parameters of derivative function part are shared cross all the derivative stages[2].

**Training Configuration.** One NVIDIA GTX 2080Ti GPU is used for training. In the experiments, all the variant DCM networks share the same training setting. The patch size is $100 \times 100$, and the batch size is 4. The ADAM algorithm is adopted to train the models with an initial learning rate $1 \times 10^{-3}$, and ends after 100 epochs. When reaching 30, 50 and 80 epochs, the learning rate is decayed by multiplying 0.2 and $\lambda$ is set as 1.0 in loss function.

**Experiment Analysis.** As shown in Table 3 and Table 4, we can clearly find that, by integrating with our proposed framwork, all the baselines obtain performance gain in image deraining and image deblurring task over standard benchmarks, which validates the effectiveness of our framework. In addition, we also report the visual results in Figure 2 and Figure 3 for image deraining and deblurring. Owing to the 3-order performs the best, it is chose as the baseline framework for implementation. Taking the trade-off 3-order DCM for example, we evaluate it over image deraining task and analyze the different high-order output of derivative function part to testify the underlying mechanism.

### 4.4 Limitations and Discussions

First, we evaluate the effectiveness of proposed framework over two typical image restoration tasks (*i.e.*, image deraining and image deblurring) and we will conduct more comprehensive experiments on restoration tasks (*e.g.*, image super-resolution, image denoising and image dehazing). Second, the focus of this work is not designing a new image restoration network, so both the mapping function and derivative function steps can be readily replaced by other advanced embodiments for better performance. since we the mapping function part is implemented only by several convolution layers in our work. It should be constructed by various complex neural architectures for comparison in the future. In addition, the parameters of derivative step are shared across different orders. We will explore the situation when the parameters are not sharing weights.

---

[2]We use two convolutional layers in the experiments, and this two-layer structure can be readily replaced by other advanced embodiments for better performance.

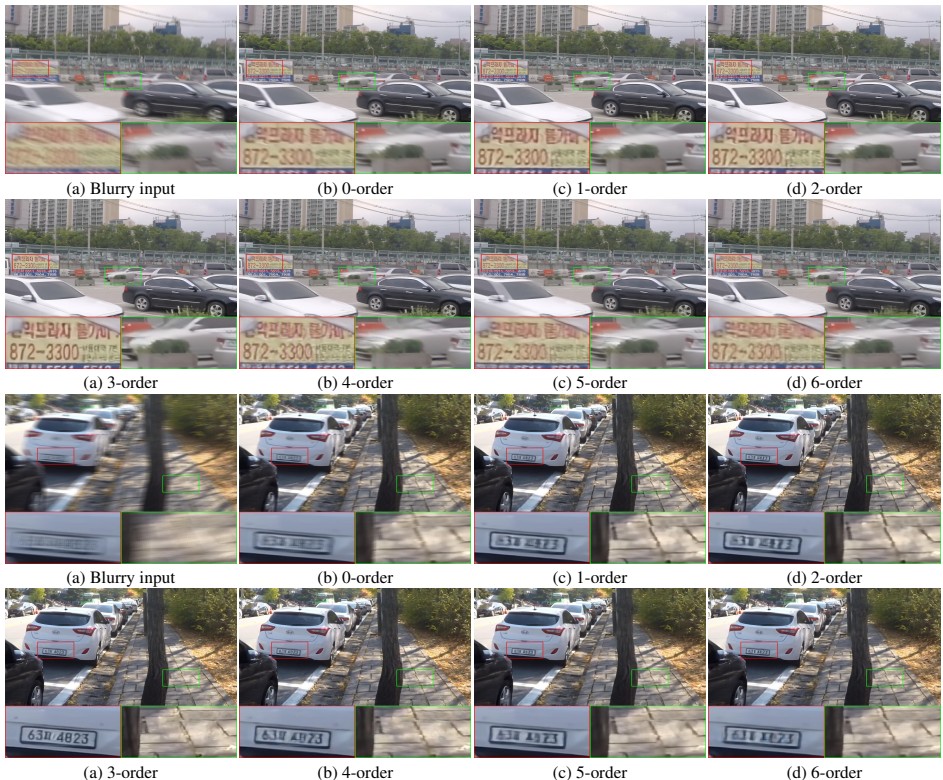

**Figure 3:** Visual results from the GoPro blurry dataset for DCM [14] with different-order framework.

## 5 Conclusion

In this paper, we propose a novel framework, Deep Taylor's Approximations Framework for image restoration by unfolding Taylor's Formula. Our proposed framework is orthogonal to existing deep learning-based image restoration methods and thus can be easily integrated with these methods for further improvement. Extensive evaluations demonstrate the effectiveness and interpretability of our framework in image restoration task, *i.e.*, image deraining and image deblurring. We believe the Deep Taylor's Approximations Framework also has potential to advance other image/video restoration tasks, *e.g.*, image/video super-resolution and image/video denoising.

## Broader Impact

Since image degradation is the most common and inevitable phenomena in imaging systems, *e.g.*, from the point spread function of the optical system to the shaking during shooting, the image restoration technology has broad impacts and practical values in various applications. Related fields include remote sensing, medicine, astronomy, military, and civilian imaging equipment. Image restoration technology aims to recover high-quality images from given low-quality counterparts. In daily life, it can help people who cannot afford professional cameras to take photos with low-end devices. Therefore, our image restoration method based on the proposed unfolding Taylor's approximations can provide high-quality clear images to facilitate intelligent data analysis tasks in these fields.

The negative consequences may accompany image restoration technology despite the many benefits it brings. This is mainly associated with certain risks of privacy and consumer experience. For example, in media or criminal cases, the identity of certain persons will be blurred to protect privacy. In this case, image deblurring technology may reveal the personal identity, thereby compromising their privacy. In addition, some users may beautify the image before sharing it. Therefore, the use of image restoration technologies to restore the posted images may arouse users' antipathy. Furthermore, it is important to be cautious of the results of any image restoration algorithms as failures, leading to misjudgments and affecting subsequent use.

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
