# OpenReview forum: "Unfolding Taylor's Approximations for Image Restoration"
_NeurIPS.cc/2021/Conference — NeurIPS 2021 Poster_

### Official Review · Reviewer_Rxva · 2021-07-13

**Rating:** 7
**Confidence:** 5

**Summary:**

This paper aims to develop an image restoration framework through a combination of Taylor’s Approximation and deep learning to break the entire recovery process into two manageable steps. Specifically, the former learns high-level contextualized information, and the latter combines it with the degraded input to progressively recover local high-order spatial details. Several promising methods are integrated with the proposed framework and achieve some improvement. Ablation studies about Taylor’s order are also conducted to provide a comprehensive analysis.

**Limitations And Societal Impact:**

The author has clearly presented and solved the limitation and potential negative societal impact.

**Main Review:**

Pros:
+ The authors innovatively propose a new image restoration framework, in which others researchers or users are likely to utilize it for further restoration performance improvement. The exploration of combining classic methods with deep learning for image restoration is worth trying and encouraging.

+ The claim of this submission is well supported by theoretical analysis. The authors revisit the connection between image restoration process and Taylor’s Formula, and break the restoration process into two manageable steps for the deep network design.

+ Extensive analysis on two image restoration tasks (image de-blurring and de-raining) clearly verify the effectiveness and scalability of the proposed method.

Cons:
-Limited reported image restoration tasks. I am curious that whether the proposed method can deal with other ill-posed image restoration tasks, like the classical super-resolution and image de-noising. It would be better if the authors could provide more discussion about it.

-Some detailed comparison between the original baselines and that integrating with Taylor’s framework is missing, e.g., the computational time and Flops.

-The premise of Taylor’s Formula is that \epsilon in Equation (6) should be small enough. Are there any other image restoration tasks that do not meet this premise?

-What is the difference between the proposed Taylor’s approximation framework and other deep unfolding frameworks, such as HQS [*1, *2] or proximal gradient operator [*3]?

[*1] Learning Deep Priors for Image Dehazing, ICCV, 2019.

[*2] Deep Plug-and-Play Super-Resolution for Arbitrary Blur Kernels, CVPR, 2019.

[*3] Deep Gradient Projection Networks for Pan-sharpening, CVPR, 2021.


**Time Spent Reviewing:**

10

---

> ### Author Response · Authors · 2021-08-08
> **Reponses to Reviewer  Rxva**
>
>
> 	**Evaluation over other ill-posed image restoration tasks.**
>
> Due to time limitation, we have only conducted the experiments over additional representative image dehazing task. The promising AOD-net [*1] is employed as the baseline on indoor hazy dataset [*2] and outdoor hazy dataset [*2] for evaluation by PSNR and SSIM. As reported bellow, the corresponding results also testify the effectiveness of our proposed Taylor’s framework. We will add more experimental results in a revision.
>
>        ----------------------------------------------------------------------------------------
>                      method               Indoor                     Outdoor
>
>                      AOD-net            18.98/0.849	            20.21/0.876
>
>                      Integrated         19.15/0.850	            20.36/0.877
>         ----------------------------------------------------------------------------------------
>
>
> [*1] AOD-NET：An All-in-One Network for Dehazing and Beyond, ICCV, 2017.
>
> [*2] Benchmarking Single Image Dehazing and Beyond, TIP, 2019.
>
> 	**The computational time and Flops comparison between the original baselines and that integrating with Taylor’s framework.**
>
> In our work, compared with the original baseline, the integrated version has only two-layer convolution with less parameters. The computational time and Flops difference can be almost negligible. Take the DCM for example, the time difference is negligible 0.8ms.
>
> 	**Which image restoration tasks do not meet this premise?**
>
> If the value of additional noise is very close to the original image pixel, this kind of image restoration may be not suitable for our proposed framework. Referring to the premise of Taylor’s Approximation, the N in Eq. (4), remarked as noise in our work, should be far less than the original image pixel by principle. Once not established, the approximation error is much larger.
>
>      **The difference between the proposed Taylor’s approximation framework and other deep unfolding frameworks.**
>
> The unfolding methods unroll an iterative algorithm involving a CNN-based regularizer by multiple concatenation stages. In each stage, the output of them is similar. Different from them, we are the first attempt to integrate the Taylor’s Approximation into image restoration network design. The different stage output has the K-order difference. By stage-wise adding the details that corresponds to the K-order output, the network result is progressively closed to the ground truth. This makes our proposed Taylor’s framework more interpretable.

---

> ### Comment · Reviewer_Rxva · 2021-08-28
> **My final score**
>
> After reading the response and other viewers' comments, I will keep my score. The authors' response solved my concerns. I think this paper indeed provides some important insights for image restoration. Looking forward to seeing the final version of this paper.

---

### Official Review · Reviewer_qVNs · 2021-07-14

**Rating:** 8
**Confidence:** 5

**Summary:**

The author proposes a novel deep neural network, which leverages the intrinsic knowledge of low-level vision and Taylor’s Approximations, for the image restoration tasks. The author found that the main part and derivative part of Taylor’s Approximations takes the internal similarity as the high-level structure and spatial details of image restoration, respectively. From this, the author breaks down the restoration process into two manageable steps, corresponding to the mapping function and derivative function, respectively. The proposed framework is orthogonal and can be easily integrated into existing deep learning-based methods. The idea of using Taylor’s Approximations to guide deep network design for image restoration is quiteThe limitation and potential negative societal impact of their work have been discussed in their paper. interesting.

**Limitations And Societal Impact:**

The limitation and potential negative societal impact of their work have been discussed in their paper.

**Main Review:**

Strengths:

1)	The author proposes a novel image restoration framework by leveraging the intrinsic knowledge of image restoration task and Taylor’s Approximations. This idea is interesting and clearly differs from previous related methods.

2)	The framework is designed by unfolding Taylor’s formula, which is sufficiently supported by theoretical analysis. This design is more reasonable than current roughly layer-stacking deep networks for image restoration.

3)	The proposed framework can be easily integrated with existing methods, which improves their performance by higher PSNR values and sharper details. Moreover, the submission is clearly written, well organized and easy-to-follow.

4)	Extensive experiments over the representative image de-blurring and image de-raining tasks are reported. Quantitative and qualitative results show its effectiveness and scalability. Besides, ablation studies about Taylor’s order are well designed and reported to analyze the order selection.

Weaknesses:

1)	The author employs Taylor’s Approximation to unfold the universal image restoration model in Line (130-132). However, it is inadequate that only image de-raining and image de-blurring tasks are reported to demonstrate the effect. More comparisons and results of other image restoration tasks, e.g., super-resolution, should be presented.

2)	The author presents the ablation experiments about Taylor’s orders. The results show that that of Order = 3 performs the best. However, the authors have not clearly illustrated the underlying reasons. For example, feature visualization or other technologies should be adopted to do so.

3)	If the loss is supervised at each layer of the derivative function part, will the effect be better?

4)	How is the generalization ability of the model in real scenes?


**Time Spent Reviewing:**

3

---

> ### Author Response · Authors · 2021-08-08
> **Responses to Reviewer qVNs**
>
>
> 	**About presenting more comparisons and results of other image restoration tasks.**
>
> Due to time limitation, we have only conducted the experiments over additional representative image dehazing task. The promising AOD-net [*1] is employed as the baseline on indoor hazy dataset [*2] and outdoor hazy dataset [*2] for evaluation by PSNR and SSIM. As reported bellow, the corresponding results also testify the effectiveness of our proposed Taylor’s framework. We will add more experimental results in a revision.
>
>               --------------------------------------------------------------------
>                   method               Indoor                    Outdoor
>
>                   AOD-net           18.98/0.849	            20.21/0.876
>
>                  Integrated         19.15/0.850	            20.36/0.877
>
>               --------------------------------------------------------------------
>
> [*1] AOD-NET：An All-in-One Network for Dehazing and Beyond, ICCV, 2017.
>
> [*2] Benchmarking Single Image Dehazing and Beyond, TIP, 2019.
>
> 	**About the reason why the 3-Order performs the best.**
>
> In Line 230-231 and 250-251, the derivate part is implemented by two convolution layers and shared across all the derivative stages. For the ablation study in Table 3 and Table 4, it can be seen that the 3-order performs the best. As for the 1-2 order, the lower results may be owing to relatively weak representative learning of two convolution layers. The higher order results is worse than the 3-order, which may be attributed to the difficult gradient backward from multiple shared concatenation of derivative part.
>
> 	**About supervision at each layer of the derivative function part.**
>
> Taking DCM as an example, we have conducted the experiments about loss supervision at each layer. In the setting where the derivate part is implemented by two convolution layers and shared across all the derivative stages, the gradient backward is difficult to optimize from multiple shared concatenation of derivative part. It results in that the performance has no obvious change between the loss supervision over each stage and current setting in our work. In addition, we also replace the current derivative part with more complex structures without sharing weights. By doing so, the results have been improved to a certain extent between loss supervision at each stage and current setting.
>
>   **The generalization ability of the model in real scenes.**
>
> In our paper, we test our framework over the real-world scenes in image deblurring task. The corresponding results are reported in Table 2. In Line 206-207, we clarify that we train the baselines over GoPro dataset and directly test it over the real-world deblurring dataset, e.g., RealBlur-J and RealBlur-R. The corresponding results in Table 2 validate the better generalization ability. Moreover, we also testify the baselines over real-world rainy dataset SPA-data. It also shows that our method has a better generalization ability.

---

> ### Comment · Reviewer_qVNs · 2021-08-18
> **After rebuttal**
>
> I have seen the authors addressed my concerns very well. In fact, those concerns are minor ones. I have come across R1's comments. I think R1's comments somehow make sense. But indeed, it is very hard for modern DNN to provide strict analysis. Since in general image restoration involves too many practical cases, I do not think providing a very strict analysis is possible. However, providing a new perspective to understand image restoration would be a new idea such as using Taylor approximation. Overall, I think this paper has its merits. I thus give the score 8.

---

### Official Review · Reviewer_FEEp · 2021-07-16

**Rating:** 6
**Confidence:** 4

**Summary:**

The authors proposed to use additional network structures to approximate the derivative parts of Taylor expansion for image restoration. The idea is similar to residual learning by recursively adding details to the images for a better visual quality.

**Limitations And Societal Impact:**

See the above comments.

**Main Review:**

Pros,

- The novel idea of combining Taylor formula with image restoration tasks, and good explanation of the motivations.
- Extensive experiments on different image restoration tasks including deraining and deblurring.

Cons,

- The proposed methods may not be as effective as expected, since the implementation of the derivative parts are the simple additional RNN structures with feature aggregation from different levels. The derivative functions are also approximated by simple convolutional layers.
- The quantitative comparisons with other baselines after integrating the new frameworks may not be fair enough, since the new model contains additional parameters and it will bring performance improvement.
- The authors claim a better detail restoration, but it cannot be reflected using PSNR or SSIM metrics. PSNR can be still higher even though the results are blur.
- The visual results are not clearly revealing the better quality, and the 3-order network is not clearly better than 0-order in visual.
- The authors empirically chose 3-order as the best-performance baseline, but what should be the theory support that 3-order network works the best? Table 3 and 4 also show some randomness of different orders, without showing clear evidence of increasing performance of using higher orders. Intuitively, it should be the case that using higher-order will bring back more details. But the results make sense given the current implementation.
- Some typos in line 250-251.

I like the idea of approximate a Taylor formula for image restoration, but the current implementation is not the best way to achieve this idea. The experiment results lack fairness, and more visual comparisons should be provided. Currently, I sit on borderline.

**Time Spent Reviewing:**

0.5

---

> ### Author Response · Authors · 2021-08-08
> **Responses to Reviewer FEEp**
>
>
> 	**About the implementation of the derivative parts.**
>
> First, we emphasize that this is the first attempt to integrate the Taylor’s Approximation with the image restoration task. Therefore, the current implementation may be not the best way to achieve this idea. Second, in the limitation and discussion part of our work, we clarify that the focus of this work is not to design a network, but a novel framework. Therefore, we only implement the derivative part with the simple two-layer convolution to testify its effectiveness. In line 249, we argue that replacing the derivative part with other advanced embodiments will improve their performance. We have conducted the above experiments as bellow by promising Residual Channel Attention Networks. Taking DCM as an example over Rain100L, the integrated DCM with Taylor’s Framework obtains 0.21 dB gain than original DCM. In future work, we will explore more effective implementation.
>
> 	**About additional parameters vs performance gain.**
>
> Taking DCM as an example, for a fair comparison, we conduct the experiments between (the DCM integrated with Taylor’s Framework) and (the DCM concatenated by the derivative part in multiple order times directly) to keep the number of parameter consistent. We adopt the 3-order for evaluation. It can be seen that the former outperforms the latter by 1.57 dB and 1.21 dB over Rain100L and Rain100H, respectively. The corresponding results clearly testify that the performance gain is from the effectiveness of our proposed Taylor’s Framework. We will add it in a revision. In addition, we also replace the current derivate part with other structures and other orders. All the experimental results also take the same effect as above.
>
>             ----------------------------------------------------------------------------------
>                            method                              Rain100L             Rain100H
>
>              DCM concatenated by the derivative part         35.74/0.975           27.56/0.868
>
>                 DCM with Taylor’s Framework                  37.31/0.980	       28.77/0.901
>             ---------------------------------------------------------------------------------
>
> 	**About visual quality measurement.**
>
> Note that the derivative part only has two-layer convolution, and it may be almost negligible compared with main baselines. Therefore, the visual quality measure is not obviously different. We have conducted the experiments by replacing it with more complex structures, like promising Residual Channel Attention Networks. Both the corresponding results and visual quality obtain improvement to some extent. Specifically, the PSNR has achieved 0.21 dB gain for DCM baseline over Rain100L. Furthermore, in the figure 3, we can clearly find that the license plate number of the 3-order is more clear than the 0-order. In addition, we will adopt more suitable measurement metric for evaluation except PSNR and SSIM as well as testify the results over downstream computer vision tasks where the small visual quality difference will seriously degrade their performance.
>
> 	**About the reason why the 3-Order performs the best.**
>
> In Line 230-231 and 250-251, the derivate part is implemented by the simple two-layer convolution and shared across all the derivative stages. For the ablation study in Table 3 and Table 4, it can be seen that the 3-order performs the best. As for the 1-2 order, the lower results may be owing to relatively weak representative learning of two convolution layers which are reused by 1-2 times. In the 4-5-6 order, the results obtained by the higher order is worse than the 3-order, which may be attributed to the difficult gradient backward from multiple shared concatenation of derivative part. Therefore, in our current implementation, the 3-order performs the best. In addition, we also replace the current derivative part with more complex structures in no sharing weights. The higher order network obtains better performance at the cost of computation burden.

---

> ### Comment · Reviewer_FEEp · 2021-08-31
> **Thanks**
>
> I have read the rebuttal. The authors address some of my concerns including parameter size difference by adding more make-up experiments. The authors claimed they replace the implementation with better module, while that will change the results of the table in the original version. I am not sure whether it is feasible. The network order problem is not well-addressed, and the 3-order option is not clearly intuitive and greatly better than other options.
>
> But the idea of the framework is still promising. I do expect future work inspired by this work to explore more on the implementation of derivation parts by adding more constraints to the module output.
>
> I will keep my rating unchanged.

---

### Official Review · Reviewer_79vC · 2021-07-16

**Rating:** 4
**Confidence:** 5

**Summary:**

This paper introduces a new perspective for designing image restoration framework by unfolding the Taylor’s Formula. The proposed framework consists of two steps, and can be integrated to existing deep networks.

**Ethics Review Area:**

["I don’t know"]

**Limitations And Societal Impact:**

Mathematically, using Taylor's formula requires several assumpations about the function, which may not be suitable for all restoration tasks.
I think it would be better to be more strict and analyze what kind of restoration tasks are suitble for this method.

**Main Review:**

Strengths
1. The authors analyze image restoration tasks in a framework of Taylor's Formula. The idea is interesting.
2. The proposed method can be integrated into exisiting methods and boost performance.

Weakness.
1. My main concern is the technical correctness part of the method.
    First, most image restoration tasks are ill-posed problems, thus a analytical and accurate inverse process is impossible by principle, as in Eq (4).
    Second, Taylor's formula is only suitble for functions that are high-order smooth. As for CNN, operators such as dropout or max-pooling are not smooth. Then what kind of CNNs are suible for the proposed method

2. How to supervised G, so its result can approximate k order output ?
    If we cannot ensure the meaning of G's output, is it just some kind of residual learning?
    I would like to see more comparisons between residual learning method and this method.

3. G is a network with additional parameters. When it is integreted to other methods, the total parameters increase.
    Is this the reason for performance gain ?

**Time Spent Reviewing:**

1

---

> ### Author Response · Authors · 2021-08-08
> **Responses to Reviewer 79vC**
>
>
> 	**About the technical correctness part.**
>
> We agree with the reviewer that obtaining an analytical and accurate inverse process is difficult. Most image restoration methods therefore attempt to optimize the practical solutions with approximation strategy. The image degradation used in our paper, i.e., Eq. (4), represents a theoretically ideal inverse process. This also has been widely adopted in previous works as an effective approximation strategy for such ill-posed problems [*1-*9]. As our focus is to revisit the connection between Taylor’s Approximation and image restoration process, we therefore follow this commonly used approximation strategy as denoted in Eq. (4) by implementing it with CNN in this paper.  As suggested by the reviewer that it is practically impossible by principle, we think that modifying Eq. (4) by replacing “=” with “≈” will make it more appropriate, and we will revise it in a revision.
>
> As for which kind of CNNs suits for our proposed framework, we first agree with the reviewer that Taylor's formula is suitable for functions that are high-order smooth. However, in current setting of our work, no baselines include the operations like max-pooling and dropout that are not smooth. Therefore, our proposed framework is only suitable for most CNN-based image restoration methods. We will clarify it in a revision.
>
> [*1] Simultaneous Fidelity and Regularization Learning for Image Restoration, TPAMI, 2021.
>
> [*2] Denoising Prior Driven Deep Neural Network for Image Restoration, TPAMI, 2019.
>
> [*3] Unfolding recurrence by Green’s functions for optimized reservoir computing, NIPS, 2020.
>
> [*4] Unfolding the Alternating Optimization for Blind Super Resolution, NIPS, 2020.
>
> [*5] LAPAR: Linearly-Assembled Pixel-Adaptive Regression Network for Single Image Super-resolution and Beyond, NIPS, 2020.
>
> [*6] Learning Deep Priors for Image Dehazing, ICCV, 2019.
>
> [*7] Deep Plug-and-Play Super-Resolution for Arbitrary Blur Kernels, CVPR, 2019.
>
> [*8] ODE-Inspired Network Design for Single Image Super-Resolution, CVPR, 2019.
>
> [*9] Jiaya Jia:  Mathematical Models and Practical Solvers for Uniform Motion Deblurring (in Motion Deblurring: Algorithms and Systems), ISBN: 9781107044364, 2014.
>
>
> 	**About approximating k-order output.**
>
> As for approximating k-order output, inspired by other model-based image restoration methods [*1-*9], we also supervise the ground truth at the output of each stage (named the order stage in our work). The proposed framework is constructed by analyzing and implicitly forcing the k-order output. It seems like the above model-based unfolding methods and both of them implicitly enables the network to output the expected result. In addition, following current CNN-based analysis, we will visualize the output of each order stage and the residual map for further validation. We will add it in a revision.
>
> Residual learning aims to learn the identity mapping and avoid the gradient vanish/explosion issue. Different from it, our proposed framework is customized for the specific image restoration task by revisiting the similarity between the Taylor’s Approximation and degradation model. Furthermore, as suggested, we conduct the experiments between residual learning and our proposed framework. To keep the number of parameters consistent for a fair comparison, we take DCM as an example and conduct the experiments between (the DCM integrated with Taylor’s Framework) and (the DCM directly concatenated by the derivative part in multiple order times). The derivative part is implemented by the residual structure as it is one of the most commonly used network modules. We abbreviate them as bellow. It can be seen that the results of the latter also are worse than the former, which testifying the effectiveness of our proposed framework.
>
> Annotation：
>
> DCM-C-R:      DCM concatenated by the derivative part with residual learning
>
> DCM-T-no-R:   DCM with Taylor’s derivative part without residual learning
>
> DCM-T-R:      DCM with Taylor’s derivative part with residual learning
>
>                 -----------------------------------------------------------------------
>                            method            Rain100L             Rain100H
>
>                           DCM-C-R             37.24                 28.71
>
>                          DCM-T-no-R           37.31	             28.77
>
>                          DCM-T-R              37.36                 28.83
>                 -----------------------------------------------------------------------
>
> 	**About additional parameters vs performance gain.**
>
> Taking DCM as an example, for a fair comparison, we conduct the experiments between (the DCM integrated with Taylor’s Framework) and (the DCM concatenated by the derivative part in multiple order times directly) to keep the number of parameter consistent. We adopt the 3-order for evaluation. It can be seen that the former outperforms the latter by 1.57 dB and 1.21 dB over Rain100L and Rain100H, respectively. The corresponding results clearly testify that the performance gain is from the effectiveness of our proposed Taylor’s Framework. We will add it in a revision. In addition, we also replace the current derivate part with other structures and other orders. All the experimental results also take the same effect as above.
>
>          -----------------------------------------------------------------------------------
>                            method                       Rain100L             Rain100H
>
>          DCM concatenated by the derivative part      35.74/0.975           27.56/0.868
>
>                  DCM with Taylor’s Framework          37.31/0.980	       28.77/0.901
>         -----------------------------------------------------------------------------------

---

### Decision · Program_Chairs · 2021-09-27

**Decision:**

Accept (Poster)

**Comment:**

This paper proposes a new method for image restoration using deep neural networks based on decomposing the problem using a Taylor approximation. 3 out of 4 reviewers appreciate the contribution as useful and novel. The most negative reviewer (79vC) has the reasonable concern that the proposed method seems mostly empirical and is not well supported by theory. Unfortunately this reviewer only provided a short review and did not engage in committee discussions or acknowledge the author response: for this reason we cannot give it too much weight in making a decision. The other reviewers acknowledged the author response and are satisfied by it. No serious concerns were identified that would invalidate the contribution made by this paper. The author response looks convincing to me as well: Authors, please integrate this discussion and all promised new results into the camera ready version of the paper.